# Systematic Evaluation of Parameters Important for Production of Native Toxin A and Toxin B from *Clostridioides difficile*

**DOI:** 10.3390/toxins13040240

**Published:** 2021-03-27

**Authors:** Aria Aminzadeh, René Jørgensen

**Affiliations:** 1Department of Bacteria, Parasites and Fungi, Statens Serum Institut, 2300 Copenhagen, Denmark; aram@ssi.dk; 2Department of Chemistry, University of Copenhagen, 2100 Copenhagen, Denmark; 3Department of Science and Environment, University of Roskilde, 4000 Roskilde, Denmark

**Keywords:** *Clostridioides difficile*, *Clostridium difficile*, TcdA, Toxin A, TcdB, Toxin B, toxin expression, protein purification, sandwich ELISA, culture media

## Abstract

In the attempt to improve the purification yield of native toxin A (TcdA) and toxin B (TcdB) from *Clostridioides difficile (C. difficile)*, we systematically evaluated culture parameters for their influence on toxin production. In this study, we showed that culturing *C. difficile* in a tryptone-yeast extract medium buffered in PBS (pH 7.5) that contained 5 mM ZnCl_2_ and 10 mM glucose supported the highest TcdB production, measured by the sandwich ELISA. These culture conditions were scalable into 5 L and 15 L dialysis tube cultures, and we were able to reach a TcdB concentration of 29.5 µg/mL of culture. Furthermore, we established a purification protocol for TcdA and TcdB using FPLC column chromatography, reaching purities of >99% for both toxins with a yield around 25% relative to the starting material. Finally, by screening the melting temperatures of TcdA and TcdB in various buffer conditions using differential scanning fluorimetry, we found optimal conditions for improving the protein stability during storage. The results of this study present a complete protocol for obtaining high amounts of highly purified native TcdA and TcdB from *C. difficile.*

## 1. Introduction

*Clostridioides difficile* (*C. difficile*) is a Gram-positive, anaerobic, and spore-forming bacterium known to be the leading cause of healthcare-associated diarrhea. *C. difficile* infection (CDI) is responsible for close to half a million incidences and 29,000 deaths annually in the Unites States alone [1,2] while in Europe the burden of CDI in acute care hospitals is estimated to be 123,997 incidences annually [3]. CDI particularly affects patients above 65 years undergoing antibiotic therapy and gives rise to a spectrum of disease symptoms, ranging from milder symptoms like fever, nausea, and diarrhea to pseudomembranous colitis, toxic megacolon, and death [4,5,6]. The first line of treatment for CDI consists of antibiotics such as metronidazole and vancomycin [7]; however, 20–30% of patients experience a recurrent infection [8].

Pathogenicity of *C. difficile* is mainly due to two secreted clostridial toxins, TcdA and TcdB, which are large proteins with molecular weights of 308 and 270 kDa, respectively, sharing 68% structural similarity [9]. Both toxins enter the host cell cytoplasm by receptor-mediated endocytosis, where the acidic pH of the endosome triggers changes in the protein conformation. These changes lead to pore formation and translocation of a catalytic glucosyltransferase domain across the endosomal membrane and into the cytosol. Once inside, the glucosyltransferase domain glucosylates and thereby inactivates small GTPases of the Rho family, causing degradation of the actin cytoskeleton, leading to apoptotic cell death [10,11]. The potency and cytotoxic effects of TcdA and TcdB are well studied, and despite some contradicting reports regarding their individual roles [12,13,14,15], most in vivo studies suggest that both toxins contribute to the severity of disease symptoms during CDI [10,12,16]. However, there is still a lot we do not know about the function of these toxins. The genes encoding TcdA (*tcdA*) and TcdB (*tcdB*) are located in a 19.6 kb region of the chromosome called the pathogenicity locus (PaLoc) [17,18]. As the genes are highly homologous, they are likely evolved by duplication. In vitro TcdA and TcdB are expressed simultaneously during the stationary growth phase in a ratio between 2:1 and 3:1 [19,20], respectively, and they are usually secreted between 16 and 72 h of growth [21]. Several studies in both humans and animals have shown that immunization with detoxified TcdA and TcdB can be used to protect against CDI symptoms [22,23,24], and several toxoid-based vaccine candidates made it to clinical trials [25,26,27]. Hence, the neutralization of TcdA and TcdB by toxin-specific antibodies is potentially an efficient method for preventing disease symptoms [28,29]. This is further highlighted by the FDA-approved TcdB specific monoclonal antibody (Bezlotoxumab) that can be used as a treatment against recurrent CDI [30]. The expression and purification of *C. difficile* toxins is therefore interesting for many research purposes.

The expression of *C. difficile* toxins is reported to be under the influence of several environmental and dietary factors [10]. Different growth media compositions have been studied and compared in regard to toxin production. Previously, brain heart infusion (BHI) culture medium was found to be the best for toxin production [31], whereas a more recent study found that the soy-based peptone medium N-Z-Soy BL4 maximized the production of toxin [32]. Rapidly metabolizable sugars such as glucose, mannitol, and fructose were shown to significantly inhibit toxin production in complex media [33], likely via carbon catabolite repression [34]. Lately, it was reported that hypervirulent *C. difficile* strains can metabolize low concentrations of trehalose; this resulted in higher disease severity and TcdB production when mice were given trehalose in their drinking water prior to a CDI challenge [35]. Other studies found that the addition of the branched-chain amino acids (BCAAs), i.e., isoleucine, leucine, and valine, have a negative effect on toxin production in complex media [36,37], in that they enhance the binding of the transcriptional regulator CodY [38]. However, another study by Ikeda et al. [39] reported that the three BCAAs actually increased toxin production in a defined medium. In the recent years, several studies reported that excess dietary zinc alters the gut microbiota and decreases the resistance to the infection, thereby exacerbating the severity of disease [40,41]. *C. difficile* quorum signaling is another factor shown to significantly influence toxin production, where it was suggested that a small (<1000 Da) thiolactone accumulates extracellularly during high cell densities and stimulates elevated toxin expression [21].

As a result of the above many diverse findings, we found a need for a comprehensive and systematic study that investigates the most reported environmental factors influencing *C. difficile* toxin production, as presented in the literature. We therefore decided to systematically study which conditions stimulate *C. difficile* to produce high amounts of toxin, as this information not only can be valuable to manufacturers depending on native *C. difficile* toxins, but also for academic researchers seeking to study the toxins in vitro as well as understand how dietary factors might affect toxin production. The latter part can potentially have clinical importance, since a better understanding of the correlation between dietary factors and toxin production can play a key role in determining the susceptibility to CDI and the severity of disease symptoms. Finally, we screen for the best buffer conditions for storage of the toxins and optimize the purification protocol, resulting in highly pure and active toxins.

## 2. Results

### 2.1. Comparison of Culture Media for Toxin Production

First, we developed and optimized an in-house sandwich ELISA assay, which is sensitive and reproducible at measuring the TcdB concentration directly in filtered crude supernatant from *C. difficile* cultures. Unfortunately, we were not able to develop an equally sensitive and reproducible assay for TcdA detection even after testing several poly- and monoclonal antibodies, and therefore we proceeded with only measuring the TcdB concentration in this study.

Initially, we tested five different well-known *C. difficile* culture media, which were all previously shown in different studies to induce high levels of TcdA and TcdB production. We also included tryptone from two different manufacturers in similar TY media for comparison. The TY^for^ medium (Formedium tryptone) supported the highest production of TcdB (1046 ng/mL) followed by N-Z-Soy BL4 (723 ng/mL), BHI (588 ng/mL), TY^bac^ (Bacto tryptone) (453 ng/mL) and lastly PY medium (192 ng/mL), which showed the poorest toxin yield compared to the others (Table 1). TY^for^ was chosen as the preferred medium to be used for further optimization in this study. Furthermore, we measured the optical density at 600 nm (OD_600_) of all the cultures and did not find a correlation between growth level and toxin production.

### 2.2. Sodium Thioglycolate and Cysteine Decreasing Toxin Production

Next, we tested whether 1 g/L of sodium thioglycolate or 0.5 g/L of cysteine added to the TY^for^ culture medium have an effect on toxin production. These two reducing agents are used in selective *C. difficile* culture media as they lower the redox potential and thereby support the anaerobic conditions required by *C. difficile*. However, we found that both reducing agents had a significant inhibiting effect on toxin production (Table 2) with a decreased toxin yield around 4-fold and 8-fold for sodium thioglycolate and cysteine, respectively, compared to cultures in TY^for^ medium only. Sodium thioglycolate also inhibited the growth of *C. difficile* whereas cysteine did not.

### 2.3. Branched-Chain Amino Acids (BCAAs) Isoleucine, Leucine, and Valine Decreasing Toxin Production

Using the TY^for^ culture medium, we then tested how an addition of equivalent molar amounts of the three BCAAs isoleucine, leucine, and valine from 10 to 100 mM affect toxin production (Table 3). We found that all concentrations of the BCAAs decreased toxin production, with 100 mM BCAA (181 ng/mL) decreasing more than 5-fold compared to the control (1067 ng/mL). BCAA concentrations of 10 mM (622 ng/mL), 25 mM (766 ng/mL), and 40 mM (630 ng/mL) were not notably different from each other in regard to toxin production but were all lower than the control.

### 2.4. Phosphate-Buffered TY^for^ Medium at pH 7.5 Increasing Toxin Production

All of the abovementioned media solutions were prepared in deionized water and the pH adjusted to 7.0 ± 0.1. Therefore, we decided to determine whether buffering the TY^for^ culture medium with either 100 mM PBS or 100 mM PBS, 5 g/L bicarbonate, both at pH 7.0, would increase toxin production (Table 4). The results indicate that PBS-buffered TY^for^ (1112 ng/mL) was slightly better at inducing toxin production compared to pH adjusted TY^for^ (943 ng/mL). However, when bicarbonate was added to the PBS-buffered TY^for^, toxin production decreased by around 3-fold to 305 ng/mL. Although the increase in toxin production when buffering TY^for^ with PBS compared to water was not significant, future experiments were carried out with PBS-buffered TY^for^.

Next, we studied toxin production in cultures of PBS-buffered TY^for^ with pH ranging from 6.5 to 8.0 (Table 5). Toxin production in PBS-buffered TY^for^ was highest at pH 7.5 (1350 ng/mL), followed by pH 7.0 (1160 ng/mL), pH 8.0 (849 ng/mL), and lastly at pH 6.5 (594 ng/mL). Hence, we continued our experiments with media solutions, with their pH adjusted to 7.5.

### 2.5. Addition of Stationary-Phase Culture Supernatant Having No Effect on Total Toxin Production

It was shown in previous studies that *C. difficile* toxin production is regulated by quorum signaling molecules that accumulate as bacterial density increases, and that adding a partially purified cell-free stationary-phase culture supernatant (from 16 to 32 h incubation) to fresh *C. difficile* cultures can induce early toxin production within 4 h of incubation [19,21]. To evaluate whether this early toxin production would affect the total toxin titer in TY^for^ medium, we boiled and filtered (at 0.22 µm) a supernatant from 24 h incubated cultures, as described by Darkoh et al. [21] before adding the supernatant (33.3%, *v/v*) to fresh *C. difficile* cultures. After 5 days of incubation, we measured the toxin concentration in cultures with or without a stationary-phase culture supernatant added (Table 6), and we found no significant difference in the toxin yield when adding the stationary-phase culture supernatant (946 ng/mL) compared to the control (1024 ng/mL). Hence, our results failed to show that the early induction of toxin production, as shown in previous studies, accumulates into a total increase of toxin after 5 days of growth.

### 2.6. Carbon Sources Added to TY^for^ Culture Medium Decreasing Toxin Production

We also studied the effect of glucose, mannitol, fructose, and trehalose in the TY^for^ culture medium. As previously shown, glucose had a negative effect on the toxin production, which could be seen even at a concentration of 5 mM glucose (914 ng/mL) (Table 7). At 10 mM glucose (649 ng/mL), the toxin production was 2-fold lower than the control (1300 ng/mL), and at 25 mM glucose (51 ng/mL) and 50 mM glucose (91 ng/mL) it was decreased by 25-fold and 14-fold, respectively. Mannitol at 10 mM (632 ng/mL) also showed around a 2-fold decrease in toxin production, whereas fructose (105 ng/mL) had a strong toxin repressing effect with a 12-fold decrease, when 10 mM was added to the medium. The addition of 10 mM trehalose (1259 ng/mL) had neither a negative nor a positive effect on toxin production.

### 2.7. Zinc Upregulating Toxin Production

We then tested the effect of zinc on toxin production and added from 1 to 10 mM ZnCl_2_ to our TY^for^ culture medium in combination with or without 10 mM glucose. Surprisingly, zinc significantly increased toxin production in a concentration-dependent manner up to 5 mM (Table 8). A ZnCl_2_ concentration of 1 mM in the TY^for^ medium stimulated a 1.5-fold increase in toxin production (1681 ng/mL) compared to the control (1078 ng/mL), and interestingly, toxin production was further increased 2.9-fold to 3155 ng/mL when 10 mM glucose was also added to the medium with 1 mM ZnCl_2_. More so, the addition of 5 mM ZnCl_2_ increased the toxin production by 2.6-fold to 2828 ng/mL compared to the control, and when combined with 10 mM glucose, it reached 4429 ng/mL, a 4.1-fold increase. Hence, at both 1 and 5 mM ZnCl_2_, the addition of 10 mM glucose further increased toxin production as a synergistic effect. The results also showed that 5 mM ZnCl_2_ was the optimal concentration, as increasing the ZnCl_2_ concentration to 10 mM in the TY^for^ medium resulted in a reduction in toxin yield (765 ng/mL), thereby having an inhibiting effect. When adding 10 mM ZnCl_2_ and 10 mM glucose in combination to the medium, the toxin production (1112 ng/mL) reached approximately the same level as the control.

We then tested the consequence of incubation time for reaching maximum toxin yield during cultivation. Samples were taken from the same cultures (TY^for^, 100 mM PBS (pH 7.5), 1 mM ZnCl_2_, 10 mM glucose) after incubation for 1, 3, and 5 days and analyzed for toxin concentration (Table 9). While there was a significant increase in toxin production from day 1 to 3, no further increase in toxin production was observed from day 3 to 5.

### 2.8. Effect of Various Metal Salts on Toxin Production

After learning that the addition of 5 mM ZnCl_2_ in combination with 10 mM glucose to the TY^for^ culture medium increased toxin production up to 4.1-fold, we studied whether other metal salts had an effect similar to ZnCl_2_ (Table 10). We chose metal salts that are commonly used in bacterial growth media, including FeSO_4_, CuSO_4_, MnCl_2_, and MgCl_2_. As seen in Table 10, the addition of 5 mM ZnCl_2_ and 10 mM glucose to the TY^for^ culture medium once more supported significantly higher toxin production (4385 ng/mL) compared to the TY^for^ control (1501 ng/mL). However, none of the other metal salts had an effect similar to ZnCl_2_. In fact, the addition of FeSO_4_ (346 ng/mL) and CuSO_4_ (10 ng/mL) both dramatically decreased toxin production, whereas MnCl_2_ (1272 ng/mL) had a slightly inhibiting effect, and MgCl_2_ (1722 ng/mL) stimulated a slightly increased toxin production.

### 2.9. Optimized Culture Medium Increasing Toxin Production at Large Scale

Up until this point, all cultivations were performed at a small scale in 100 mL Pyrex^®^ bottles containing 50 mL TY^for^ media. To routinely produce and purify toxins in larger scales, we used dialysis tubes immersed into 5 L Erlenmeyer flasks filled with TY^for^ medium, as this system is known to increase toxin expression. Seed cultures were then used to inoculate the medium inside the dialysis tubes, while the surrounding medium was kept sterile during cultivation. Therefore, we tested whether the optimized PBS (pH 7.5)-buffered TY^for^ with 5 mM ZnCl_2_ and 10 mM glucose culture medium (TY^opt^) when used in the large-scale dialysis tube cultures likewise improved the toxin production. The optimized TY^opt^ culture medium did in fact result in an increased toxin production in the large-scale dialysis tube system (Table 11). We reached a toxin concentration of 29,500 ng/mL in our optimized culture medium, which is around a 3.3-fold increase compared to previous large-scale cultures using the same dialysis tube system in TY^for^ media which only yielded 8868 ng/mL on average. The fold increase of toxin production at a small-scale could therefore be translated to large-scale cultures.

We also cultured the commonly preferred *C. difficile* strain VPI 10463 in a TY^opt^ medium at a small scale and found that this strain reached a TcdB titer similar to strain R20291 (see Appendix A).

Finally, for the ease of understanding which conditions supported an increased production of TcdB, we summarized them in Table 12.

### 2.10. Purification of C. difficile TcdA and TcdB

We optimized a purification protocol for TcdA and TcdB using column chromatography with an initial diafiltration step. The diafiltration step removed the culture medium as well as smaller contaminating proteins and nucleotides, while dialyzing the sample into a 50 mM Tris-HCl (pH 7.5) buffer. This step resulted in a significant loss of TcdB to 66.2% of the starting material (Table 13). Following the diafiltration, the material was loaded onto a Q Sepharose ion exchange column, which increased the purity of TcdA and TcdB to 34% (Appendix A) and 65.6% (Appendix A), respectively, with an accompanying loss to 43.2% of the initial TcdB material (Table 13). As the Q Sepharose column separated TcdA and TcdB, both toxins were hereafter individually purified using a high resolution Mono Q ion exchange column, with TcdA buffered in 50 mM Tris-HCl (pH 7.5), while TcdB was buffered in 50 mM Tris (pH 7.5) containing 50 mM CaCl_2_. The Mono Q ion exchange step increased the purity of TcdA and TcdB to 72.5% and 95.2%, respectively, resulting in 36% of the initial amount of TcdB remaining (Table 13). As a final purification step, we performed size exclusion chromatography using a Superdex 200 Increase 10/300 GL column to reach a purity of >99% for both toxins (Appendix A). The final yield of TcdB was 25.2% relative to the starting material, and we assume a comparable yield of TcdA since the same purification steps were used for both toxins.

### 2.11. Optimal Buffer Conditions for TcdA and TcdB

Finally, we also screened for the optimal buffer conditions in order to determine the highest protein stability during storage; to do this, we used differential scanning fluorimetry (DSF) to determine the melting temperature (*T*_m_) of TcdA and TcdB. First, we used a RUBIC buffer screen (Molecular Dimensions, Sheffield, SYK, United Kingdom) to screen for a range of diverse buffers commonly used in structural biology, including pH, salts, and different concentrations [42]. We followed this by a RUBIC additive screen (Molecular Dimensions, Sheffield, SYK, United Kingdom) to screen for small molecules that can affect protein folding, solubility, and stabilization. The buffer, pH, salts, and additive that showed the highest *T*_m_ for each toxin in the RUBIC screens were selected and then further analyzed. The optimal buffer and pH for TcdA was found to be 100 mM HEPES and pH 7.5, respectively, whereas for TcdB it was 100 mM HEPES and pH 7.0, respectively. Hereafter, as the buffer and pH were kept constant, optimal salts and additives selected from the RUBIC screens were added in different concentrations to screen for the final optimal conditions (Appendix A). The optimal buffer condition for TcdA reaching a *T*_m_ of 56.7 °C was found to be 100 mM HEPES (pH 7.5), 500 mM MgSO_4_, and 500 mM glutamic acid (Table 14). Similarly, for TcdB the optimal buffer condition reaching a *T*_m_ of 53.1 °C was 100 mM HEPES (pH 7.0), 250 mM NaSO_4_, and 250 mM glutamic acid (Table 14). These optimized storage buffer conditions markedly increased the *T*_m_ values and protein stability of both toxins compared to the 50 mM Tris-HCl (pH 7.5) that we previously used.

## 3. Discussion

Prompted by a low yield of TcdA and TcdB when culturing *C. difficile* in BHI media, we decided to systematically screen for conditions to optimize the toxin production. We therefore systematically evaluated a range of different parameters that were previously shown to affect native toxin production in *C. difficile*, including various media compositions, carbon sources, metal salts, additives, and growth conditions, for their ability to induce toxin production.

First, we evaluated different well-known culture media in comparison to BHI, which is a well-established culture medium used since the 1980s for *C. difficile* toxin production [43,44]. Our BHI culture medium was prepared slightly differently than the standard recipe, since we additionally added 20 g/L yeast extract so that we can compare it directly to the other culture media also containing similar amounts of yeast extract. We found that the tryptone-based culture medium that contains tryptone manufactured by Formedium (TY^for^) consistently supported the highest toxin production among the five different media tested, reaching a concentration of 1046 ng/mL TcdB after 5 days of incubation (Table 1). Interestingly, a similar culture medium containing Bacto tryptone from BD Biosciences (TY^bac^) instead of Formedium reduced the toxin yield more than 2-fold to 453 ng/mL. Tryptone is a pancreatic digest of casein and is often used in combination with yeast extract (TY) as culture medium for *C. difficile* [32,33,37]. We could not find any detailed nutritional composition distinguishing the tryptones from the two different manufacturers, which could explain the dramatic difference in toxin production. BHI and PY media were both inferior to the TY^for^ medium, but more interestingly, the soy-based medium N-Z-Soy BL4 was also outperformed by the TY^for^ medium, although a previous study evaluating different culture media showed the opposite [32]. The study showed that the N-Z-Soy BL4 medium was markedly better at stimulating *C. difficile* toxin production compared to the TY medium. However, one distinguishing factor is their use of *C. difficile* strain VPI 10463, which is ribotype 087, whereas in this study we used ribotype 027 (strain R20291). The TY^for^ culture medium supporting the highest toxin production was used as a control medium as we further evaluated different parameters throughout the study.

The two reducing agents, sodium thioglycolate and cysteine, both decreased the toxin production (Table 2). Sodium thioglycolate at 1 g/L dramatically inhibited bacterial growth, which could explain the resulting lower yield of toxin. Cysteine, however, had no effect on the growth as the OD_600_ values were approximately the same regardless of cysteine addition, but the addition of 0.5 g/L cysteine resulted in an 8-fold decrease in toxin yield. An earlier study found a similar result, where cysteine was a potent downregulator of metabolic pathways including toxin production and more than 30 other proteins in *C. difficile* in both PY and BHI media [45]. Cysteine and/or sodium thioglycolate are commonly used to maintain low redox conditions during *C. difficile* cultivation [46,47], which was our rationale for using them in this study, but we concluded that the proposed benefit of maintaining low redox conditions by adding these reducing agents to the medium did not outweigh the dramatic decrease in toxin production.

The three BCAAs of isoleucine, leucine, and valine all had a poor effect on toxin production when added in conjunction to the TY^for^ medium at various concentrations from 10 to 100 mM, with the highest concentration by far being the poorest (Table 3). Thus, these results support previous studies showing that the three BCAAs are among nine amino acids (Cys, Gly, Ile, Leu, Met, Pro, Tyr, Val, and Trp) that all have a negative effect on toxin production when added to rich media [35,36], while only in a defined medium can they upregulate the production of toxin, probably due to them being essential for growth and not by directly affecting toxin expression [39,48].

We further found that buffering the TY^for^ medium with 100 mM PBS was slightly better at inducing toxin production compared to TY^for^ mixed in deionized water (Table 4). The increased toxin production by using PBS was not significant, however we decided to continue buffering the medium with PBS and possibly prevent the previously reported acidification of the medium during cultivation [49]. Interestingly, we found that adding 5 g/L bicarbonate to the PBS-buffered TY^for^ medium had a dramatic negative effect on the toxin yield with a 3-fold decrease, without affecting the growth. This is in contrast to a previous study showing that adding bicarbonate to PY culture media buffered with PBS increased the levels by 10-fold [36]. We also found that the pH of the medium plays a role, as maintaining a neutral pH between 7.0 and 7.5 (with 7.5 being slightly better than 7.0) was markedly better compared to a slightly acidic or alkaline pH of 6.5 or 8, respectively (Table 5). These results are supported by a previous study showing that TcdB production was highest at pH 7.5 when cultivating the *C. difficile* strain VPI 10463 in liquid Gifu Anaerobic Medium ranging from pH 4.5 to 8.5 [50]. In contrast, there is a recent paper by Wetzel and McBride et al. [49], which found that the *C. difficile* strain R20291 (same strain as the one in this study) produced the highest amount of TcdA at pH 5.5 (measured by western blot) when cultivated on solid media, more than 2-fold higher than at pH 7.0 and 7.5. However, the findings of the study showing that pH 5.5 was superior at supporting toxin production likely is due to the cultivation on solid media known to induce high levels of sporulation, since sporulation and toxin expression are known to be coregulated in *C. difficile* [51,52,53]. We tested *C. difficile* cultivation in TY^for^ medium at pH 5 and found almost undetectable TcdB levels (data not shown).

*C. difficile* quorum signaling is known to positively regulate toxin production, by means of a small (<1000 Da) extracellular thiolactone signaling molecule accumulating during high cell densities as suggested by Darkoh et al. [21]. The study found that by adding the cell-free stationary-phase culture supernatant from high-density *C. difficile* cells to a fresh culture, it was possible to induce early toxin production after only 4 h rather than 16 h. However, it was not clear from the study whether the early toxin production also leads to a total increase in accumulated toxin over time. Therefore, we tested this hypothesis and demonstrated that cultures not exposed to the stationary-phase culture supernatant reached the same toxin levels after 5 days of incubation, possibly because the stationary-phase supernatant only stimulates premature toxin production and not a total increase in toxin titers (Table 6).

Next, we found that the rapidly metabolizable sugars (glucose, mannitol, and fructose) had a negative effect on the toxin production (Table 7). Concurrently, it was shown that the inhibiting effect of glucose on toxin expression was in a concentration-dependent manner, starting even at a concentration as low as 5 mM. It is well-known from the literature that these sugars will inhibit toxin production in *C. difficile* via carbon catabolite repression [34]. However, we wanted to explore whether low concentrations could stimulate higher cell densities without directly suppressing toxin production, which leads to higher levels of secreted toxin. In the same experiment, we found that trehalose did not have the same suppressing effect on toxin production, but neither did it increase it. Furthermore, we tested a higher trehalose concentration in the TY^for^ medium, showing decreased toxin titers, and we also tested low to high trehalose concentrations in a basal defined medium, showing similar decreased titers as for the TY^for^ medium (data not shown). These results are in contrast to the in vivo study by Collins et al. [35], showing that hypervirulent *C. difficile* strains, while having the ability to metabolize low concentrations of trehalose, will increase TcdB expression, resulting in higher disease severity in mice that were given trehalose in the drinking water. The authors extrapolated from those results and discussed that the increased dietary trehalose consumption by humans since the early 2000s likely contributed to the spread of epidemic *C. difficile* ribotypes. However, our data do not support the conclusion that trehalose directly increases TcdB production in the hypervirulent *C. difficile* strain R20291.

A surprising finding is the dramatic effect of zinc on toxin production, as the addition of 1 or 5 mM ZnCl_2_ to the TY^for^ medium increased the toxin production to 1681 ng/mL and 2828 ng/mL, respectively (Table 8). An even more interesting finding is the consistent synergistic effect of combining ZnCl_2_ with 10 mM glucose, as this further upregulated the toxin expression to 3155 ng/mL and 4429 ng/mL for 1 mM and 5 mM ZnCl_2_, respectively. We cannot explain this synergistic effect of glucose and ZnCl_2_ on toxin production, as glucose alone suppresses toxin production in the same medium (Table 7). The reason for choosing zinc was based on the studies by Zackular et al. [40] and Zackular and Skaar [41], which demonstrated that dietary zinc affects CDI severity in mice. The authors discussed that excess dietary zinc alters the gut microbiota and decreases resistance to CDI, and while this might also be the case, our study clearly shows that the reason for increased CDI severity could be a direct effect by zinc on the toxin production of *C. difficile.* Further investigation into other metal salts including FeSO_4_, CuSO_4_, MnCl_2_, and MgCl_2_ did not show the same effect as for ZnCl_2_ (Table 10). This shows that zinc specifically, has an active role in supporting toxin production of *C. difficile* in our optimized TY^opt^ medium. All the conditions showing increased toxin production are summarized in Table 12. We also found that incubating *C. difficile* cultures for more than 3 days did not secrete more toxin (Table 9); instead, we often experience that incubation at 37 °C beyond 3 days will risk degradation of the already secreted toxin as well as contamination of the culture.

By reaching a 4.1-fold increase in concentration of native TcdB in small-scale 50 mL cultures, we decided to evaluate our optimized medium conditions in large-scale 5 L cultures using immersed dialysis tubes. We found that our TY^opt^ culture medium supported TcdB production up to 29,500 ng/mL, which is a 3.3-fold increase compared to previous dialysis tube cultures using nonoptimized TY^for^ medium (Table 11). Since then, we further scaled up the dialysis tube cultures into 15 L Nalgene^TM^ culture vessels (Thermo Fisher Scientific, Waltham, MA, USA), and we were able to reproduce the same toxin titers as the 5 L cultures (data not shown), demonstrating that our improved culture media conditions for toxin production are scalable.

The purification of *C. difficile* toxins was described extensively throughout several decades, and the most common purification protocol consists of using ammonium sulfate precipitation, followed by ion exchange chromatography to separate TcdA and TcdB [43,54,55]. To our knowledge, no studies have systematically described the purification process of TcdA and TcdB in regard to the protein loss during each purification step and the final yield of toxin relative to the starting material. We avoided ammonium sulfate precipitation as it would become a tedious bottleneck when working with larger volumes, so instead we used diafiltration to exchange the culture medium to a suitable buffer and remove smaller contaminant proteins, nucleotides, ions, and other small molecules. Previously, to prepare for column chromatography, we used 6–8 kDa cut-off dialysis tubing; however, we found this step to be severely time-consuming and insufficient in removing smaller media proteins, making the subsequent purification steps more tedious. The diafiltration step resulted in a significant loss of the TcdB pool, likely due to toxin particles being pushed through the 50 kDa membrane. We were able to reduce this loss by running the permeate through the diafiltration system a second time (data not shown). The following Q Sepharose and Mono Q ion exchange steps separated TcdA and TcdB and significantly increased purity (Table 13). Adding 50 mM CaCl_2_ to the running and elution buffer of TcdB for the Mono Q purification step efficiently removed highly persistent low molecular weight bands seen on SDS-PAGE (Appendix A), which was also previously shown in [56]_._ A final size exclusion chromatography was necessary to remove what is likely persistent nucleotides remaining in the TcdB pool prior to this step, which caused a high A_260_/A_280_ ratio. Our purification protocol resulted in a final TcdB yield of around 25% relative to the starting material and with purities of >99% for TcdB (Appendix A) and TcdA (Appendix A), respectively. As mentioned previously, our ELISA assay was inefficient at measuring the concentration of TcdA from crude culture supernatant. However, by measuring the final amount of highly purified TcdA from our large-scale cultures using a NanoDrop ND-1000 spectrophotometer, we observe that the ratio between purified TcdA and TcdB is approximately 2:1. Previous studies have reported ratios of TcdA and TcdB to be between 2:1 [19] and 3:1 [20], which is in accordance with our own results from extensive experience with purification studies.

Finally, we improved the buffer conditions for the storage of TcdA and TcdB by screening various buffers, salts, pH, and additives using DSF to find the conditions supporting the highest melting temperatures. Before the screening, we used a common 50 mM Tris-HCl buffer (pH 7.5) to store the toxins, and this condition was previously shown to support *T*_m_ values of 51.5 °C and 49 °C for TcdA and TcdB, respectively [22]. The improved buffer conditions for TcdA were found to be 100 mM HEPES (pH 7.5), 500 mM MgSO_4_, 500 mM glutamic acid with a *T*_m_ of 56.7 °C, while for TcdB, the optimal buffer conditions were 100 mM HEPES (pH 7.0), 250 mM NaSO_4_, 250 mM glutamic acid with a *T*_m_ of 53.1 °C (Table 14). We therefore see a substantial increase in thermal stability when storing the toxins in the respective optimized buffer conditions.

## 4. Conclusions

In conclusion, our study revealed that a culture medium based on tryptone and yeast extract, supplemented with zinc and glucose, supported the highest *C. difficile* toxin production, consistently reaching 4-fold higher toxin levels than previous culture media. Interestingly, the addition of zinc to the culture medium caused the most dramatic influence on toxin production. Furthermore, we find that glucose, usually toxin-suppressing, had the opposite effect when in the presence of zinc, where it stimulated increased toxin production. These results indicate that zinc and glucose in combination can directly exacerbate the severity of *C. difficile* pathogenesis, supporting previous studies that found excess dietary zinc as a cause of decreased resistance to CDI [40,41]. In addition, we found that trehalose did not have any effect on TcdB production, opposing an earlier in vivo study by Collins et al. [35], where mice showed higher disease severity and TcdB expression after being given trehalose in the drinking water. Finally, we identified the optimal buffer conditions and established a purification protocol for TcdA and TcdB, achieving more than 99% purity.

## 5. Materials and Methods

### 5.1. Chemicals and Reagents

Tryptone, yeast extract, and HEPES were obtained from Formedium (Hunstanton, NK, UK), while Difco^TM^ Bacto tryptone was obtained from BD Biosciences (San Jose, CA, USA). N-Z-Soy BL4, peptone, sodium thioglycolate, l-cysteine, l-isoleucine, l-leucine, l-valine, Trizma base, NaCl, SYPRO orange dye, sodium bicarbonate, d-(+)-glucose, d-(−)-fructose, d-(+)-trehalose dihydrate, zinc chloride (ZnCl_2_), iron(II) sulfate heptahydrate (FeSO_4_ 7H_2_O), copper(II) sulfate pentahydrate (CuSO_4_·5H_2_O), magnesium chloride hexahydrate (MgCl_2_ 6H_2_O), and manganese(II) chloride tetrahydrate (MnCl_2_ 4H_2_O), H_2_SO_4_, Na_2_CO_3_, NaHCO_3_, bovine serum albumin (BSA) and glycerol (>99%) were purchased from Sigma-Aldrich (St. Louis, MO, USA). d-(−)-mannitol, Tween^®^ 20 and PEG6000 was obtained from Merck Chemicals GmbH (Darmstadt, HE, Germany). Oxoid^TM^ Brain Heart Infusion (BHI) broth was purchased from Thermo Fisher Scientific (Waltham, MA, USA). Polyclonal rabbit anti-TcdB (catalog code: pAK a-TcdB) and monoclonal mouse anti-TcdB (catalog code: MAb-B71) and (catalog code: MAb-B72) antibodies were obtained from tgcBIOMICS (Bingen, RP, Germany). HRP-conjugated rabbit anti-mouse (H+L) antibody (Cat #6170-05) was purchased from Southern Biotech (Birmingham, AL, USA). TMB PLUS2 was obtained from Kem-En-Tec Diagnostics A/S (Taastrup, Denmark).

### 5.2. Culture Media

The TY^for^ (tryptone-yeast extract) culture medium contained 30 g/L tryptone (Formedium) and 20 g/L yeast extract. The TY^bac^ (Bacto tryptone-yeast extract) culture medium contained 30 g/L Bacto tryptone (BD Biosciences), and 20 g/L yeast extract. N-Z-Soy BL4 culture medium contained 30 g/L N-Z-Soy BL4 and 20 g/L yeast extract. BHI culture medium contained 37 g/L brain heart infusion broth and 20 g/L yeast extract. The PY (peptone-yeast extract) culture medium contained 30 g/L peptone and 20 g/L yeast extract. The composition for TY seed medium used in this study was as follows: 24 g/L tryptone (Formedium) and 12 g/L yeast extract.

### 5.3. Microorganism

The organism used for all toxin production studies was *C. difficile* strain R20291 (NCTC 13366), purchased from Public Health England (Salisbury, WL, UK). In a single comparative experiment (Appendix A), *C. difficile* strain VPI 10463 (ATCC 43255), purchased from LGC Standards (Teddington, MX, UK), was used. The freeze-dried cultures were resuspended in TY medium and incubated in an Oxoid^TM^ 3.5 L Anaerobic jar (Thermo Fisher Scientific, Waltham, MA, USA) with Anoxomat^®^-controlled atmosphere of 96% N_2_/4% H_2_ at 37 °C (MART Microbiology B.V., Lichtenvoorde, The Netherlands). After 24 h incubation, 30% sterile glycerol was added, and the culture was divided into 1 mL cryogenic vials and stored at −80 °C until further use.

### 5.4. Anaerobic Growth

Anaerobic growth conditions (anaerobiosis) were achieved by autoclaving for 25 min at 121 °C. Immediately after autoclaving, all media were incubated in Oxoid^TM^ 3.5 L Anaerobic jars with Anoxomat^®^-controlled atmosphere of 96% N_2_/4% H_2_ at 37 °C for at least 24 h prior to inoculation with *C. difficile*.

### 5.5. Seed Culture

To prepare first-stage seed culture, 15 mL of TY seed medium was added to a 16 × 150 mm borosilicate glass tube and autoclaved for 25 min at 121 °C. Immediately after autoclaving, the first-stage seed tube was incubated under anaerobic conditions at 37 °C for at least 24 h, before being inoculated with 1 mL of *C. difficile* glycerol stock and incubated unagitated for 24 h at 37 °C under anaerobic conditions. Second-stage seed (50 mL TY seed medium) was prepared as first-stage seed but in a 100 mL Pyrex^®^ bottle, and inoculated with 0.5 mL first-stage seed culture and incubated unagitated for 24 h at 37 °C under anaerobic conditions. Cell density was measured using a SmartSpec 3000 spectrophotometer (Bio-Rad, Hercules, CA, USA) at an optical density of 600 nm (OD_600_).

### 5.6. Bacterial Cultures

Culture media were dispensed at 50 mL per each 100 mL Pyrex^®^ bottle and autoclaved for 25 min at 121 °C. After being equilibrated to anaerobic conditions at 37 °C for at least 24 h, the culture medium was inoculated with 0.5 mL second-stage seed culture and incubated unagitated under anaerobic conditions for either 3 or 5 days at 37 °C. Large-scale cultures using dialysis tubes were prepared as described previously in [22]. Briefly, a Spectra/Por^®^ 1 6-8 kDa dialysis tube (Repligen, Rancho Dominguez, CA, USA) was filled with 500 mL PBS (pH 7.5) and immersed in 4 L of TY culture medium in a 5 L Pyrex^®^ Erlenmeyer flask and autoclaved for 30 min at 121 °C. All culture media were pH-adjusted with HCl before autoclaving. After sterilization and creation of anaerobiosis, the medium was equilibrated overnight at 37 °C prior to inoculation, during which growth nutrients diffuse into the dialysis tube. Five mL of second-stage seed culture was inoculated (1%, *v/v*) into the dialysis tube and left undisturbed for 3 days at 37 °C. All bacterial cultures were repeated at least two or more times for each culture condition tested.

### 5.7. Partial Purification of Stationary-Phase Culture Supernatant

Stationary-phase culture supernatant was partially purified as described in [21]. Briefly, cultures incubated for 24 h were boiled for 20 min, centrifuged at 10,000× *g* for 10 min, and the supernatant was filtered at 0.22-μm. Hereafter, the supernatant was dialyzed using Amicon^®^ Ultra-15 30K centrifugal filters (Merck Millipore Ltd., Carrigtwohill, CO, Ireland) where the retentate containing toxins and larger proteins was discarded, and the permeate containing the quorum signaling molecules was kept. The permeate was dialyzed using a Spectra/Por^®^ Biotech CE 100–500 Da dialysis tube immersed in PBS (pH 7.5) with 30% (*w/v*) PEG 6000 to simultaneously concentrate the permeate. The partially purified and concentrated stationary-phase culture supernatant, which contained quorum signaling molecules, was added (33.3%, *v/v*) to fresh *C. difficile* cultures immediately before inoculation.

### 5.8. Measuring TcdB Concentration by Sandwich ELISA

Bacterial culture was centrifuged at 10,000× *g*, and the supernatant filtrated through a 0.22-μm syringe filter. The supernatant/filtrate was then assayed for a concentration of TcdB using the sandwich ELISA method. Polystyrene MaxiSorp microtiter plates (Nunc, Roskilde, Denmark) were coated with 100 μL/well of 1 μg/mL rabbit anti-TcdB antibody in 0.05 M Na_2_CO_3_, 0.05 M NaHCO_3_ (pH 9.6), and were incubated overnight at 5 °C. The next day, 300 μL of blocking buffer, 1% (*w/v*) BSA in PBS-0.05% (*v/v*) Tween 20 (pH 7.4) was added to the wells and incubated for at least 1 h at 37 °C. Cell-free culture supernatants, standards, and controls were all diluted in the blocking buffer, added at 100 μL/well in duplicates and incubated for 1 h at 37 °C. After this, 100 μL of the secondary antibody, i.e., mouse anti-TcdB antibody, was added at 1 μg/mL to each well and incubated for 1 h at 37 °C. Monoclonal mouse anti-TcdB (Lot#B71-39836) secondary antibody was used for measuring TcdB from *C. difficile* R20291, while monoclonal mouse anti-TcdB (Lot#B72-40017A) secondary antibody was used for measuring TcdB from *C. difficile* VPI 10463. For detection, 100 μL of HRP-conjugated rabbit anti-mouse IgG antibody, diluted 1:2500 in blocking buffer, was added to each well, followed by incubation for 1 h at 37 °C. Antibody binding was visualized by the addition of 100 μL/well TMB PLUS2 substrate and incubated at 37 °C for 10 min, after which the reaction was stopped by adding 100 μL/well of 0.2 M H_2_SO_4_. Absorbance was measured at 450 nm using a POLARstar OPTIMA microplate reader (BMG laboratories, Ortenberg, HE, Germany). Plates were washed between each step with PBS-0.05% Tween 20.

### 5.9. Purification of TcdA and TcdB

The bacterial culture in the dialysis tube was centrifuged at 18,500× *g* for 20 min at 4 °C and dialyzed using a Quattro 1000 Ultrafiltration/Diafiltration pump with Pellicon^®^ 2 Biomax 50 kDa membrane cassettes (Merck Millipore Ltd., Carrigtwohill, CO, Ireland) in 50 mM Tris-HCl (pH 7.5). Separation of TcdA and TcdB from the dialyzed supernatants was achieved using a self-packed Q Sepharose Fast Flow anion-exchange (GE Healthcare, Chicago, IL, USA) column, integrated on an Äkta Purifier FPLC (GE Healthcare, Chicago, IL, USA). The toxins were eluted with a linear 0 to 1 M NaCl gradient in 50 mM Tris-HCl (pH 7.5), with TcdA eluting between a conductivity of 9 and 18 mS/cm and TcdB between 46 and 55 mS/cm. Eluted fractions were visualized on SDS-PAGE, and protein sizes corresponding to either TcdA or TcdB were pooled. Both toxins were diluted in buffer to reduce the NaCl concentration, the pooled TcdA being diluted 5-fold in 50 mM Tris-HCl (pH 7.5), while the pooled TcdB being diluted 5-fold in 50 mM Tris-HCl (pH 7.5) containing 50 mM CaCl_2_. Both toxins were further purified using a high-resolution anion-exchange Mono Q 5/50 GL column (GE Healthcare, Chicago, IL, USA), with TcdA eluting between 8 and 22 mS/cm and TcdB eluting between 28 and 46 mS/cm. As the final step, both toxins were concentrated using Amicon^®^ Ultra-15 10K centrifugal filters and purified using a Superdex 200 Increase 10/300 GL size-exclusion column (GE Healthcare, Chicago, IL, USA).

### 5.10. Determination of Melting Temperature by Differential Scanning Fluorimetry (DSF)

Using a Applied Biosystems^®^ MicroAmp^TM^ 96-well plate (Thermo Fisher Scientific, Waltham, MA, USA), 2 μL of SYPRO Orange dye (62× concentrated stock) was mixed with 2 μL of 1.2 μM TcdA or TcdB and 23 μL of individual buffers to a final volume of 25 μL. The plate was centrifuged for 1 min at 2300× *g* before being placed into the ABI 7500 Real-Time Polymerase Chain Reaction machine (Thermo Fisher Scientific, Waltham, MA, USA). The temperature gradient was set to run from 20 to 95 °C with an increase of 1 °C/min, as described previously [42]. The fluorescence signal was recorded, and the obtained data were analyzed and processed on Prism software version 8.3.0 (GraphPad, San Diego, CA, USA, 2019).

### 5.11. Statistical Analysis

At least two or more independent biological replicates were made for each culture condition, as specified in each table. All biological replicates were measured on ELISA in technical duplicates of which the mean value was used. The TcdB concentration of each culture condition is calculated as the median value of the biological replicates, with the lowest and highest values shown in parentheses.

## Figures and Tables

**Table 1 toxins-13-00240-t001:** Toxin production in different culture media.

Culture Medium	Medium Components	pH	OD_600_	TcdB (ng/mL)
BHI	Brain Heart Infusion/yeast extract	7.0	0.79	588 (534–643) (*n* = 2)
TY^for^	Tryptone/yeast extract	7.0	0.82	1046 (1039–1054) (*n* = 2)
TY^bac^	Tryptone/yeast extract	7.0	1.36	453 (409–496) (*n* = 2)
N-Z-Soy BL4	N-Z-Soy BL4/yeast extract	7.0	0.90	723 (578–869) (*n* = 2)
PY	Peptone/yeast extract	7.0	0.81	192 (191–192) (*n* = 2)

All cultures were incubated unagitated for 5 days at 37 °C, and their optical density at 600 nm (OD_600_) was measured on day 5. ^for^ = tryptone (Formedium), ^bac^ = Bacto tryptone (BD Biosciences).

**Table 2 toxins-13-00240-t002:** Effect of sodium thioglycolate and cysteine on toxin production.

Culture Medium	pH	OD_600_	TcdB (ng/mL)
TY^for^	7.0	0.8	1094 (987–1200) (*n* = 2)
TY^for^, sodium thioglycolate	7.0	0.2	270 (263–277) (*n* = 2)
TY^for^, cysteine	7.0	0.84	129 (109–134) (*n* = 3)

Sodium thioglycolate was added at 1 g/L and cysteine was added at 0.5 g/L. All cultures were incubated unagitated for 5 days at 37 °C, and OD_600_ was measured on day 5. ^for^ = tryptone (Formedium).

**Table 3 toxins-13-00240-t003:** Effect on toxin production by BCAAs.

Culture Medium	pH	OD_600_	TcdB (ng/mL)
TY^for^	7.0	1.0	1067 (904–1200) (*n* = 4)
TY^for^, BCAA (10 mM) ^a^	7.0	0.76	622 (359–811) (*n* = 3)
TY^for^, BCAA (25 mM) ^a^	7.0	0.86	766 (568–959) (*n* = 6)
TY^for^, BCAA (40 mM) ^a^	7.0	0.91	630 (572–851) (*n* = 4)
TY^for^, BCAA (100 mM) ^a^	7.0	0.67	181 (142–221) (*n* = 2)

All cultures were incubated unagitated for 5 days at 37 °C and OD_600_ were measured on day 5. ^a^ The concentration of BCAA specifies the concentration of isoleucine, leucine, and valine added to the culture medium. ^for^ = tryptone (Formedium).

**Table 4 toxins-13-00240-t004:** Effect of buffering TY^for^ with PBS or PBS/bicarbonate on toxin production.

Culture Medium	Solvent	pH	OD_600_	TcdB (ng/mL)
TY^for^	H_2_O	7.0	0.82	943 (834–1051) (*n* = 2)
TY^for^	PBS	7.0	0.90	1112 (1001–1223) (*n* = 2)
TY^for^	PBS/bicarbonate	7.0	0.79	305 (235–375) (*n* = 2)

Bicarbonate and PBS were added at 5 g/L and 100 mM, respectively. All cultures were incubated unagitated for 5 days at 37 °C, and OD_600_ was measured on day 5. ^for^ = tryptone (Formedium).

**Table 5 toxins-13-00240-t005:** Effect of pH on toxin production.

Culture Medium	pH	OD_600_	TcdB (ng/mL)
TY^for^	6.5	0.70	594 (498–729) (*n* = 3)
TY^for^	7.0	0.78	1160 (946–1214) (*n* = 3)
TY^for^	7.5	0.85	1350 (1324–1376) (*n* = 3)
TY^for^	8.0	0.89	849 (775–922) (*n* = 3)

TY^for^ culture media were buffered with 100 mM PBS. All cultures were incubated unagitated for 5 days at 37 °C, and OD_600_ was measured on day 5. ^for^ = tryptone (Formedium).

**Table 6 toxins-13-00240-t006:** Effect on toxin production by adding stationary-phase culture supernatant.

Culture Medium	pH	OD_600_	TcdB (ng/mL)
TY^for^	7.5	0.54	1024 (1020–1095) (*n* = 3)
TY^for^, culture supernatant	7.5	0.64	946 (850–1097) (*n* = 3)

TY^for^ culture media were buffered with 100 mM PBS (pH 7.5). Stationary-phase culture supernatant was added at (33.3%, *v/v*). All cultures were incubated unagitated for 5 days at 37 °C with the OD_600_ measured on day 5. ^for^ = tryptone (Formedium).

**Table 7 toxins-13-00240-t007:** Effect of carbon sources on toxin production.

Culture Medium	pH	OD_600_	TcdB (ng/mL)
TY^for^	7.5	1.04	1300 (1250–1344) (*n* = 3)
TY^for^, 5 mM glucose	7.5	0.80	914 (773–1146) (*n* = 3)
TY^for^, 10 mM glucose	7.5	0.65	649 (505–713) (*n* = 6)
TY^for^, 25 mM glucose	7.5	0.74	51 (48–53) (*n* = 3)
TY^for^, 50 mM glucose	7.5	0.71	91 (89–102) (*n* = 3)
TY^for^, 10 mM mannitol	7.5	0.86	632 (502–655) (*n* = 3)
TY^for^, 10 mM trehalose	7.5	0.88	1259 (1226–1333) (*n* = 3)
TY^for^, 10 mM fructose	7.5	0.60	105 (84–123) (*n* = 3)

TY^for^ culture media were buffered with 100 mM PBS (pH 7.5) and incubated unagitated for 5 days at 37 °C with OD_600_ measured on day 5. ^for^ = tryptone (Formedium).

**Table 8 toxins-13-00240-t008:** Effect of zinc and glucose on toxin production

Culture Medium	pH	OD_600_	TcdB (ng/mL)
TY^for^	7.5	1.17	1078 (987–1228) (*n* = 6)
TY^for^, 1 mM ZnCl_2_	7.5	0.74	1681 (1396–1809) (*n* = 4)
TY^for^, 1 mM ZnCl_2_, 10 mM glucose	7.5	1.15	3155 (2985–3318) (*n* = 3)
TY^for^, 5 mM ZnCl_2_	7.5	0.80	2828 (2494–3154) (*n* = 6)
TY^for^, 5 mM ZnCl_2_, 10 mM glucose	7.5	0.85	4429 (3779–4433) (*n* = 3)
TY^for^, 10 mM ZnCl_2_	7.5	0.72	765 (700–891) (*n* = 3)
TY^for^, 10 mM ZnCl_2_, 10 mM glucose	7.5	0.68	1112 (1029–1596) (*n* = 3)

TY^for^ culture media were buffered with 100 mM PBS (pH 7.5), and ZnCl_2_ was filtrated into the culture flasks after media was autoclaved. All cultures were incubated unagitated for 3 days at 37 °C with OD_600_ measured on day 3. ^for^ = tryptone (Formedium).

**Table 9 toxins-13-00240-t009:** Comparison of incubation time.

Culture Medium	Incubation (Days)	pH	OD_600_	TcdB (ng/mL)
TY^for^, 1 mM ZnCl_2_, 10 mM glucose	1	7.5	1.77	1172 (1106–1398) (*n* = 3)
TY^for^, 1 mM ZnCl_2_, 10 mM glucose	3	7.5	1.15	3155 (2985–3318) (*n* = 3)
TY^for^, 1 mM ZnCl_2_, 10 mM glucose	5	7.5	0.86	3136 (3114–3340) (*n* = 3)

TY^for^ culture media were buffered with 100 mM PBS (pH 7.5), and ZnCl_2_ was filtrated into the culture flasks after media was autoclaved. All cultures were incubated unagitated for 5 days at 37 °C, while samples were taken on days 1, 3, and 5 and measured for OD_600_ and TcdB concentration. ^for^ = tryptone (Formedium).

**Table 10 toxins-13-00240-t010:** Effect of various metal salts on toxin production

Culture Medium	pH	OD_600_	TcdB (ng/mL)
TY^for^	7.5	1.19	1501 (1485–1508) (*n* = 3)
TY^for^, 5 mM ZnCl_2_, 10 mM glucose	7.5	0.84	4385 (3576–4683) (*n* = 3)
TY^for^, 5 mM FeSO_4_, 10 mM glucose	7.5	1.23	346 (327–580) (*n* = 3)
TY^for^, 5 mM CuSO_4_, 10 mM glucose	7.5	0.23	10 (9–11) (*n* = 3)
TY^for^, 5 mM MnCl_2_, 10 mM glucose	7.5	1.03	1272 (1190–1453) (*n* = 3)
TY^for^, 5 mM MgCl_2_, 10 mM glucose	7.5	0.86	1722 (1649–2059) (*n* = 3)

TY^for^ culture media were buffered with 100 mM PBS (pH 7.5) and incubated unagitated for 3 days at 37 °C with OD_600_ measured on day 3. All metal salts were filtrated into the culture flasks after media was autoclaved. ^for^ = tryptone (Formedium).

**Table 11 toxins-13-00240-t011:** Effect of medium optimization in large-scale dialysis tube cultures.

Culture Medium	Volume (mL)	pH	OD_600_	TcdB (ng/mL)
TY^for^ (flask)	50	7.5	0.82	1046 (1039–1054) ^a^ (*n* = 3)
TY^opt^ (flask)	50	7.5	0.85	4429 (3779–4433) ^b^ (*n* = 3)
TY^for^ (dialysis tube)	425	7.5	5.5	8868 (7728–10,401) (*n* = 5)
TY^opt^ (dialysis tube)	400	7.5	5.3	29,500 (28,983–34,029) (*n* = 4)

TY^for^ culture media were buffered with 100 mM PBS (pH 7.5) and incubated unagitated for 3 days at 37 °C with OD_600_ measured on day 3. ^for^ = tryptone (Formedium). ^opt^ = Optimized TY^for^ culture medium buffered with PBS (pH 7.5), 5 mM ZnCl_2_ and 10 mM glucose. ^a^ Result from Table 1. ^b^ Result from Table 8.

**Table 12 toxins-13-00240-t012:** Summary of all conditions that increased toxin production.

TcdB Expression	Condition	Fold Increase
Culture medium	Tryptone/yeast extract (TY^for^)	1
Buffering	100 mM PBS, pH 7.5	1.3
Metal salt	5 mM ZnCl_2_	2.7
Metal salt, sugar	5 mM ZnCl_2_, 10 mM glucose (TY^opt^)	4.1
Scale-up (dialysis tube)	5 L culture medium ^a^	28.2

^for^ = tryptone (Formedium). ^opt^ = Optimized TY^for^ culture medium buffered with PBS (pH 7.5), 5 mM ZnCl_2_ and 10 mM glucose. ^a^ Dialysis tube culture immersed in 5 L Pyrex^®^ Erlenmeyer flask.

**Table 13 toxins-13-00240-t013:** Protein loss during purification of TcdB.

Stage	Purification Step	Volume (mL)	TcdB		Starting Material (%) ^b^	Purity ^c^
			µg/mL	Total (mg)		
1	Supernatant	810 ^a^	17.2	13.9	100	N/A
2	Diafiltration	230 ^a^	39.8	9.2	66.2	3.9%
3	Q Sepharose	40	149.4	6.0	43.2	65.6%
4	Mono Q	18.6	266.6	5.0	36.0	95.2%
5	Ultrafiltration	0.26	19,425.9	5.0	36.0	98%
6	Superdex 200	3	1167.0	3.5	25.2	>99%

^a^ Volume containing TcdA and TcdB. Subsequent volumes contain only TcdB. ^b^ Calculated as the total amount of TcdB at each stage relative to the filtered supernatant (stage 1). ^c^ Purity from SDS-PAGE (Appendix A) was estimated using the lane profile function of Image Lab 6.1 software function on a Bio-Rad Gel Doc Imager.

**Table 14 toxins-13-00240-t014:** Optimized buffer conditions for TcdA and TcdB.

Toxin	Buffer (mM)	Salt (mM)	Additive (mM)	pH	*T*_m_ (°C)
TcdA ^a^	Tris-HCl (50)	-	-	7.5	51.5
TcdA ^b^	HEPES (100)	MgSO_4_ (500)	glutamic acid (500)	7.5	56.7
TcdB ^a^	Tris-HCl (50)	-	-	7.5	49
TcdB ^b^	HEPES (100)	NaSO_4_ (250)	glutamic acid (250)	7.0	53.1

*T*_m_ values were estimated from DSF measurements using real-time PCR with a temperature gradient from 20 to 95 °C with an increase of 1 °C/min. ^a^ Non-optimized buffer condition (before this study). ^b^ Optimized buffer conditions (during this study). - = not added.

## Data Availability

Data is contained within the article or Appendix A.

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
