# Peer review of "Systematic Evaluation of Parameters Important for Production of Native Toxin A and Toxin B from Clostridioides difficile"

_toxins, 2021, doi:10.3390/toxins13040240_

Round 1

Reviewer 1 Report

Although the manuscript is not very novel, but authors systematically tested conditions/ parameters affecting native TcdB production in C diff culture. This information is useful for the field.

The major weakness is that authors did not look at the effects of these conditions on TcdA production due to the failure to quantitate of TcdA by ELISA.  I would suggest  authors to overcome the issue of measuring TcdA to include the effects on different conditions on TcdA production. In fact, not much much to be done.

Author Response

Regarding the suggestion of also measuring the concentration of TcdA in different culture conditions. We agree with this suggestion and have already extensively tried to set up a sandwich ELISA sensitive to the detection of TcdA in the crude culture supernatant.

We have tried numerous combinations of 6 different monoclonal anti-TcdA antibodies (from tgcBIOMICS) and 3 different polyclonal anti-TcdA (from tgcBIOMICS and BIO-RAD) in developing a sandwich ELISA. Unfortunately, the ELISA absorbance when detecting TcdA in the culture supernatant was low and did not consistently follow a “dose-response” correlation.

We believe the reason is that the polyclonal anti-TcdA antibodies are not sufficiently sensitive to be used as ELISA plate coating antibody in this assay. Furthermore, all commercially available monoclonal anti-TcdA antibodies that we could find are from mouse species, and therefore not suitable for being used as both the capture/coating and detection antibody in a sandwich ELISA.

After numerous attempts, we decided to continue our study by only measuring the TcdB concentration. This is acceptable since we know from previous publications that the ratio between expressed TcdA:TcdB is reported to be between 2:1 and 3:1 [1,2]. We also have extensive experience from our own in-house purification studies on TcdA and TcdB, where we find a ratio between TcdA:TcdB of around 2:1 when both toxins are highly purified from the same culture. This is now described in more detail in the introduction and discussion of the manuscript, please see the revised manuscript line 50 and 487-491.

Finally, assuming we actually had established a sandwich ELISA for measuring the concentration of TcdA and wanted to include this data in the manuscript. Then we would have to repeat all the culture conditions described in the manuscript, which would delay the project for several months. 

  1. Martin-Verstraete, I.; Peltier, J.; Dupuy, B. The regulatory networks that control Clostridium difficile toxin synthesis. Toxins (Basel). 2016, 8, 1–24, doi:10.3390/toxins8050153.
  2. Hundsberger, T.; Braun, V.; Weidmann, M.; Leukel, P.; Sauerborn, M.; Von Eichel-Streiber, C. Transcription analysis of the genes tcdA-E of the pathogenicity locus of Clostridium difficile. Eur. J. Biochem. 1997, 244, 735–742, doi:10.1111/j.1432-1033.1997.t01-1-00735.x.

Reviewer 2 Report

The manuscript entitled "Systematic evaluation of parameters important for production of native toxin A and toxin B from Clostridioides difficile" presents a systematic evaluation of the influence of various culture parameters on toxin production.

While there are several previous studies that have dealt with this issue, the major strength of the current study is the simultaneous evaluation of large number of various parameters (such as sugars, amino acids, etc.).

As the authors stated, this subject is important both for manufacturers and academic researchers. In my opinion, it is also important from a clinical view, since some of the dietary may affect disease severity. I would have added the clinical importance to the manuscript. 

Specific comments:

Methods:

5.2 Did you try Thioglycollate broth medium?

5.3 I would have used several different strains of C. difficile or at least several different isolates of the tested strain, in order to confirm the presented effects of the various parameters. Additionally, the use of one strain, as presented in the current study,  would not allow a statistical analysis

5.7 Why did not you use a commercial ELISA kit for the measurements of toxins? (for example, ELISA kit of TGCbiomics)

Results:

2.5 The purpose of this experiment was not clear, since to best of my knowledge, boiling of the cell-free supernatant would inactivate the toxins. I was expected to a comparison between cultures with and without the cell-free culture supernatant.

* Overall, there is a large amount of data and too many tables. I recommend on better organizing of results, maybe addition of a summarizing diagram.

Discussion

Overall, the discussion is good. However, again, as with the results, there is too much data and it is hard to follow. Again, I recommend using a summarizing diagram.

Conclusions

I recommend adding the clinical importance of this study. This way, the study can attract more readers.  

Author Response

Specific comments: (See author's reply in blue)

Methods:

5.2 Did you try Thioglycollate broth medium?

We did not test thioglycolate broth medium as suggested. However, we tested the effect of adding 1 g/L of sodium thioglycolate to the Tryptone-yeast extract medium and saw that it significantly decreased the growth of C. difficile and thereby also the toxin production (see Results chapter 2.2). The idea was to test if the reducing effects of sodium thioglycolate would create a more anaerobic environment and support higher toxin production. This was not the case in our study, and therefore we discontinued the use.

5.3 I would have used several different strains of C. difficile or at least several different isolates of the tested strain, in order to confirm the presented effects of the various parameters. Additionally, the use of one strain, as presented in the current study,  would not allow a statistical analysis

We have tested two different reference strains in this study; a ribotype 027 strain R20291 (NCTC 13366) as our main strain, and also a ribotype 087 strain VPI10463 (ATCC 43255) for comparison. We saw a similar increased level of toxin production in both of these strains as shown in “Supplementary Table S1”. This is also described in the manuscript, lines; 263-265 and 556-558.

5.7 Why did not you use a commercial ELISA kit for the measurements of toxins? (for example, ELISA kit of TGCbiomics)

Initially, we used the commercial ELISA kit from tgcBIOMICS, but these are quite expensive, around 500€ incl. shipping for one single plate of 96 wells. We have used >50 ELISA plates for this study, so the cost of commercial ELISA kits would have accumulated significantly. Instead, we purchased monoclonal and polyclonal anti-toxin antibodies from tgcBIOMICS and developed our own sandwich ELISA. We thought of two benefits by doing this. The first benefit was to save money for buying commercial kits, and the second was to describe a sensitive and optimized sandwich ELISA assay for the benefit of other academic researchers to use for detecting TcdB in the crude culture supernatant.

Results:

2.5 The purpose of this experiment was not clear, since to best of my knowledge, boiling of the cell-free supernatant would inactivate the toxins. I was expected to a comparison between cultures with and without the cell-free culture supernatant.

It is correct that the purpose of this experiment was to compare freshly inoculated C. difficile cultures growing either with or without the addition of stationary-phase culture supernatant from a previously grown culture. We wanted to study the effect of adding culture supernatant containing quorum signaling molecules to fresh C. difficile cultures. The experiment was performed exactly as described by Darkoh et al. [1] The only difference was that we measured the TcdB concentration after 5 days and not 4 hours as they did, as our main interest was to study the final accumulated toxin production.

To further clarify, we are not boiling the culture supernatants in which we measure the TcdB concentration (Table 6). We only boil and sterile filter the partially purified culture supernatant from the previous stationary-phase culture which had been incubated for 24 h. The purpose of boiling this supernatant before adding it to the fresh cultures is that we want to denature any residual toxins being transferred from the previous cultures into the fresh cultures, as we only want to study the effect of the quorum signaling molecules, which are temperature resistant as shown by Darkoh et al. [1].

We have elaborated on the method used for this experiment and added a new chapter to the Methods section “5.7”, see the revised manuscript, line 592-603.

* Overall, there is a large amount of data and too many tables. I recommend on better organizing of results, maybe addition of a summarizing diagram.

We agree that the amount of data is large as we have performed a vast number of culture studies, and we have therefore added a new summarizing table in the Results chapter (Table 12), please see the revised manuscript line 271-273. 

Discussion

Overall, the discussion is good. However, again, as with the results, there is too much data and it is hard to follow. Again, I recommend using a summarizing diagram.

We have added a summarizing table to the Results chapter (Table 12), please see the revised manuscript line 271-273. 

Conclusions

I recommend adding the clinical importance of this study. This way, the study can attract more readers.  

We agree with the reviewer that the clinical importance of this study can attract more readers. Therefore, we have revised the Introduction and Conclusion chapters of the manuscript to broaden the target group, please see the revised manuscript, lines; 85-88 and 504-511.

We would also like to highlight sections in the Discussion chapter where we discuss the potential dietary effects in vivo, please see lines; 420-434 and 435-446.

1.    Darkoh, C.; Dupont, H.L.; Norris, S.J.; Kaplan, H.B. Toxin synthesis by Clostridium difficile is regulated through quorum signaling. MBio 2015, 6, 1–10, doi:10.1128/mBio.02569-14.

Round 2

Reviewer 1 Report

Authors responded appropriately to my questions